# Stylized Dialogue Generation with Multi-Pass Dual Learning

**Jinpeng Li**[1], **Yingce Xia**[2], **Rui Yan**[3,5*], **Hongda Sun**[3], **Dongyan Zhao**[1,4*], **Tie-Yan Liu**[2]

[1]Wangxuan Institute of Computer Technology, Peking University, Beijing, China
[2]Microsoft Research Asia, Beijing, China
[3]Gaoling School of Artificial Intelligence, Renmin University of China, Beijing, China
[4]Institute for Artificial Intelligence, Peking University, Beijing, China
[5]Beijing Academy of Artificial Intelligence, Beijing, China
lijinpeng@stu.pku.edu.cn, {yingce.xia, tyliu}@microsoft.com,
{ruiyan, sunhongda98}@ruc.edu.cn, zhaody@pku.edu.cn

## Abstract

Stylized dialogue generation, which aims to generate a given-style response for an input context, plays a vital role in intelligent dialogue systems. Considering there is no parallel data between the contexts and the responses of target style $S_1$, existing works mainly use back translation to generate stylized synthetic data for training, where the data about context, target style $S_1$ and an intermediate style $S_0$ is used. However, the interaction among these texts is not fully exploited, and the pseudo contexts are not adequately modeled. To overcome the above difficulties, we propose multi-pass dual learning (MPDL), which leverages the duality among the context, response of style $S_1$ and response of style $S_0$. MPDL builds mappings among the above three domains, where the context should be reconstructed by the MPDL framework, and the reconstruction error is used as the training signal. To evaluate the quality of synthetic data, we also introduce discriminators that effectively measure how a pseudo sequence matches the specific domain, and the evaluation result is used as the weight for that data. Evaluation results indicate that our method obtains significant improvement over previous baselines.

## 1 Introduction

In recent years, dialogue systems have attracted a surge of research interest and achieved great success [1, 2, 3, 4]. The dialogue generation task can be formulated as a sequence-to-sequence learning problem [5, 6], where the source sequence is the given context, and the target sequence is the response. Stylized dialogue generation, which automatically generates response following the desired style, obtains significant progress in open-domain dialogue systems. The styles could be formal/informal, modern/Shakespearean, serious/humor, etc.

A core challenge of this task is the lack of parallel training data between the contexts and responses of desired styles (denoted as style $S_1$), which limits the performance of models. A natural solution is to establish the pseudo pairs between contexts and responses of style $S_1$ using unsupervised or semi-supervised learning. The parallel data between contexts and responses of style $S_0$ (denoted as style $S_0$) is leveraged. Note the conversations corpus (context, style $S_0$) are relatively easy to obtain. After that, the stylized dialogue model can be trained on the synthetic data. Specifically, Su et al. [7] propose a diversifying dialogue generation model based on iterative back translation [8]. Zheng et al. [9] propose a style routing approach with a joint training process, where an inverse model with style embeddings is used to generate pseudo pairs and train the stylized dialogue generation model.

---

*corresponding authors: Rui Yan (ruiyan@ruc.edu.cn) and Dongyan Zhao (zhaody@pku.edu.cn).

35th Conference on Neural Information Processing Systems (NeurIPS 2021).

Although the above methods achieved success in the stylized dialogue generation, there still exists lots of room to improve. First, the interaction between context, style $S_0$ and style $S_1$ is not fully exploited. An example is that a pseudo context can be explicitly obtained by a two-hop process, $S_1 \rightarrow S_0 \rightarrow$ pseudo context instead of using style embedding only. Second, in the aforementioned methods, the quality of the pseudo contexts generated through back translation is not properly modeled. By manually checking the generated data, we find that there are quite a few pseudo contexts that contain irrelevant contents of the response. Several common patterns in the pseudo contexts and responses include "I don't know", "I'm so happy", etc. In previous methods, each pseudo data is treated equally regardless of their quality, which introduces bias to the training set and hurts the performances of model.

In this paper, we propose a new framework, multi-pass dual learning (MPDL), for stylized dialogue generation. Inspired by dual learning [10] that can automatically extract information through two dual tasks in an unsupervised manner, we extend it to a multi-pass version. In MPDL, there are three dual tasks in total, including context↔style $S_1$, context↔style $S_0$ and style $S_0$ ↔style $S_1$. The unlabeled texts in style $S_1$ and contexts are leveraged. We also introduce two discriminators to evaluate the quality of the generated data. The discriminators are used to measure the similarity of generated data to the corresponding domains. The similarity is used as the weight of the synthetic data for training, by which we can adaptively use the unlabeled data. We conduct experiments with benchmarks of stylized response generation in formal English and Shakespearean English. We achieved state-of-the-art results in terms of both automatic evaluation and manual judgment.

Our contributions are summarized as follows:

(1) We propose multi-pass dual learning (MPDL) framework for stylized dialogue response generation, that can effectively leverage the unlabeled data; (2) Compared with standard dual learning, we introduce two discriminators to evaluate the quality of the pseudo parallel data. This is a new attempt for the general dual learning framework; (3) We provide a new dataset for this task and set several benchmarks using our method; (4) We empirically verify the effectiveness of MPDL on two datasets with formal and Shakespearean response generation.

## 2   Related work

**Text style transfer** aims to re-write a sentence with given stylistic properties. Some early works try supervised approaches, where a work is the transfer between modern English and Shakespearean style sentences [11, 12]. Nevertheless, more and more recent research begins to pay attention to the unsupervised approach due to the lack of parallel corpora [13, 14]. In general, there are three typical approaches. The first is to learn disentangled representations of content and style. Shen et al. [15] learned a shared representation of context across different styles and then leveraged the refined alignment of latent representations for style transfer. Fu et al. [16] utilize a multi-decoder with a style embedding model to learn separate content representations, where style representations are obtained via adversarial networks. Another is to generate stylized texts with back translation [17, 18], which produces pseudo parallel data to jointly train the forward and backward model. The third is based on the template, which tries to replace original stylistic words with the target stylistic words [19, 14].

**Stylized dialogue generation** refers to generate a response in the target style. Zhou et al. [20] propose an Emotional Chatting Machine (ECM) model for the emotional response. An intuitive solution generates a response with a pre-trained style language model [21], which builds a response generation model STYLEDGPT on top of a pre-trained language model DialoGPT [22] and devise both a word-level loss and a sentence-level loss to fine-tune the DialoGPT towards the target style. Niu and Bansal [23] propose three weakly supervised models that can generate diverse, polite (or rude) dialogue responses without parallel data. Some work bridge conversation modeling and non-parallel style transfer by sharing a structured latent space [24]. Recently, some work iteratively tuned to find the optimal mapping relation between conversational context and non-conversational utterances via iterative back translation [7, 9]. These works have achieved good performance, but we have achieved better diversity and efficiency through an auxiliary domain.

**Back translation** is to train a target-to-source seq2seq model for producing source sentences, and then construct pseudo parallel datasets. This approach is widely used in machine translation [8, 25], text style transfer [26, 27] and dialogue generation [7, 9]. Similar concurrent work is **Dual learning** [10], which is first proposed for neural machine translation, involves two tasks with an immediate reward.

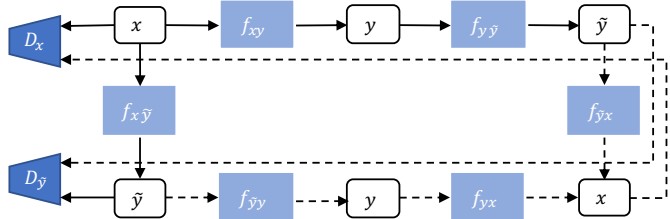

Figure 1: The framework multi-pass dual learning. $f$ denotes the encoder and decoder model constructed by GPT-2. $x$, $y$ and $\tilde{y}$ denote context, style $S_0$ and style $S_1$ respectively. $D_x$ and $D_{\tilde{y}}$ are discriminators.

Dual learning has proved to be very effective in neural machine translation and image translation [28]. Our work is also based on the idea of them. However, most of the recent works have focused on two domain mapping. We employ dual learning to work on a three-domain text related problem, then the contents of non-conversational text can be effectively utilized to enrich the dialogue generation.

# 3 Method

Let $\mathcal{C}$, $\mathbb{S}_0$, $\mathbb{S}_1$ denote the domains of context, response in style $S_0$ and response in style $S_1$ respectively. Our task is to learn a mapping, that can map a sequence from context domain $\mathcal{C}$ to a stylized response domain $\mathbb{S}_1$. A core challenge is that there is no parallel data between $\mathcal{C}$ and $\mathbb{S}_1$, but there are some related parallel corpora: (i) A dialogue dataset $\mathcal{D}_{\mathrm{dia}} = \{(x_i, y_i)\}_{i=1}^n$ where $x_i \in \mathcal{C}$ and $y_i \in \mathbb{S}_0$; (ii) A style transfer dataset $\mathcal{D}_{\mathrm{tra}} = \{(y_i, \tilde{y}_i)\}_{i=1}^m$, where $y_i \in \mathbb{S}_0$ and $\tilde{y}_i \in \mathbb{S}_1$. The corpus sizes of the above two datasets are $n$ and $m$. The sequences in style $S_0$ are not the same between $\mathcal{D}_{\mathrm{dia}}$ and $\mathcal{D}_{\mathrm{tra}}$, i.e., $\{y|(x, y) \in \mathcal{D}_{\mathrm{dia}}\} \cap \{y|(y, \tilde{y}) \in \mathcal{D}_{\mathrm{tra}}\} = \emptyset$. In addition to parallel data, there is another unlabeled stylized corpus $\mathcal{D}_{\mathrm{sty}} = \{\tilde{y}_i\}_{i=1}^l$ with size $l$, where $\tilde{y}_i \in \mathbb{S}_1$.

## 3.1 Multi-pass dual learning framework

To fully utilize the parallel and unlabeled data, we propose multi-pass dual learning (briefly, MPDL), which is illustrated in Figure 1. There are three pairs of dual tasks involved in our framework: (i) Dialogue generation and the inversed task, $f_{xy} : \mathcal{C} \mapsto \mathbb{S}_0$, $f_{yx} : \mathbb{S}_0 \mapsto \mathcal{C}$; (ii) Style transfer between $\mathbb{S}_0$ and $\mathbb{S}_1$, $f_{y\tilde{y}} : \mathbb{S}_0 \mapsto \mathbb{S}_1$, $f_{\tilde{y}y} : \mathbb{S}_1 \mapsto \mathbb{S}_0$; (iii) Stylized dialogue response generation and the inversed task, $f_{x\tilde{y}} : \mathcal{C} \mapsto \mathbb{S}_1$, $f_{\tilde{y}x} : \mathbb{S}_1 \mapsto \mathcal{C}$.

Our ultimate goal is to obtain the stylized dialogue response generation model $f_{x\tilde{y}}$, while the remaining tasks are all auxiliary tasks. For multi-pass dual learning, given any $\tilde{y} \in \mathcal{D}_{\mathrm{sty}}$, we could obtain reconstruction $\tilde{y}' = f_{x\tilde{y}}(f_{yx}(f_{\tilde{y}y}(\tilde{y})))$ by sequentially applying $f_{\tilde{y}y}$, $f_{yx}$ and $f_{x\tilde{y}}$. The reconstruction error between $\tilde{y}'$ and $\tilde{y}$ determines the ability of model reconstruction. That is, we want to maximize the logarithmic probability:

$$\log P(\tilde{y}' = \tilde{y}|\tilde{y}; f_{\tilde{y}y}, f_{yx}, f_{x\tilde{y}}). \tag{1}$$

Directly optimization the above probability is difficult since there are three models to be updated. Consequently, we derive a relaxed version of the above probability. We have that:

$$\log P(\tilde{y}' = \tilde{y}|\tilde{y}; f_{\tilde{y}y}, f_{yx}, f_{x\tilde{y}}) = \log \sum_{x \in \mathcal{C}} \sum_{y \in \mathbb{S}_0} P(\tilde{y}', y, x|\tilde{y}; f_{\tilde{y}y}, f_{yx}, f_{x\tilde{y}})$$

$$= \log \sum_{x \in \mathcal{C}} \sum_{y \in \mathbb{S}_0} P(\tilde{y}'|y, x, \tilde{y}; f_{\tilde{y}y}, f_{yx}, f_{x\tilde{y}}) P(y, x|\tilde{y}; f_{\tilde{y}y}, f_{yx}, f_{x\tilde{y}}). \tag{2}$$

Considering that $\tilde{y}'$ is only related to $x$ and $f_{x\tilde{y}}$, we have that: $P(\tilde{y}'|y, x, \tilde{y}; f_{\tilde{y}y}, f_{yx}, f_{x\tilde{y}}) = P(\tilde{y}'|x, f_{x\tilde{y}})$. Due to the concavity of $\log(\dots)$, we can convert the Eqn.(2) to:

$$\log P(\tilde{y}' = \tilde{y}|\tilde{y}; f_{\tilde{y}y}, f_{yx}, f_{x\tilde{y}}) \geq \sum_{x \in \mathcal{C}} \sum_{y \in \mathbb{S}_0} P(y, x|\tilde{y}; f_{\tilde{y}y}, f_{yx}, f_{x\tilde{y}}) \log P(\tilde{y}'|x, f_{x\tilde{y}})$$

$$= \sum_{x \in \mathcal{C}} \sum_{y \in \mathbb{S}_0} P(x|y; f_{yx}) P(y|\tilde{y}; f_{\tilde{y}y}) \log P(\tilde{y}'|x, f_{x\tilde{y}}), \tag{3}$$

where the last equation is obtained by the fact that $x$ is only related to $y$ and $f_{yx}$, and $y$ is only related to $\tilde{y}$ and $f_{\tilde{y}y}$.

Instead of maximizing $\log P(\tilde{y}' = \tilde{y}|\cdots)$ directly, we maximize a lower bound as Eqn.(3) suggests. We use a sampling-based method to solve this problem: (i) Given a $\tilde{y}$, we sample a $y$ according to $P(\cdot|\tilde{y}; f_{\tilde{y}y})$; (ii) Based on $y$, we sample a $x'$ according to $P(\cdot|y; f_{yx})$; (iii) We maximize $\log P(\tilde{y}|x'; f_{x\tilde{y}})$. We can use either beam search or random sampling to get the intermediate sequences. On the other hand, we can pick an $x$ from the set $\{x|x \in \mathcal{D}_{\text{dia}}\}$, sequentially use two sampling steps $y \sim P(\cdot|x; f_{xy})$ and $\tilde{y}' \sim P(\cdot|y; f_{y\tilde{y}})$ and then maximize $\log P(x|\tilde{y}'; f_{\tilde{y}x})$. In addition to the loss for unlabeled data, we propose loss functions on the parallel data. Therefore, there are four loss functions in total:

$$\mathcal{L}_{\text{dia}} = \mathop{\mathbb{E}}_{(x,y)\sim\mathcal{D}_{\text{dia}}} - (\log P(y|x; f_{xy}) + \log P(x|y; f_{yx})),$$

$$\mathcal{L}_{\text{tra}} = \mathop{\mathbb{E}}_{(y,\tilde{y})\sim\mathcal{D}_{\text{tra}}} - (\log P(\tilde{y}|y; f_{y\tilde{y}}) + \log P(y|\tilde{y}; f_{\tilde{y}y})), \quad (4)$$

$$\mathcal{L}_{\text{dual},1}(\tilde{y}) = -\log P(\tilde{y}|x'; f_{x\tilde{y}}),$$

$$\mathcal{L}_{\text{dual},2}(x) = -\log P(x|\tilde{y}'; f_{\tilde{y}x}),$$

where $x'$ and $\tilde{y}'$ are generated by the MPDL model as the above steps.

## 3.2 Network architecture

All the six models $f$'s are built on Transformer-based architecture [29]. The encoder and decoder of $f$'s are initialized using pre-trained DialoGPT weights trained on conversation corpus of Reddit community [22]. To reduce the memory cost, (i) the parameters of all *forward* models, including $f_{xy}$, $f_{y\tilde{y}}$ and $f_{x\tilde{y}}$ are shared; (ii) the parameters of all *backward* models, including $f_{yx}$, $f_{\tilde{y}y}$ and $f_{\tilde{y}x}$ are shared; (iii) the both encoder and decoder (excluding the encoder-decoder attention modules) share the same parameters too.

Taking the *forward* models as an example, to distinguish the models from each other, we introduce several domain embeddings $T = \{E_C, E_{s_0}, E_{s_1}\}$ for context domatin (i.e., $\mathcal{C}$), style $S_0$ (i.e., $\mathbb{S}_0$) and style $S_1$ (i.e., $\mathbb{S}_1$) domains respectively. Given an input $x$ and the domain $T$, the encoder of the forward model works as follows:

$$H_f(x) = \texttt{enc}_f(E_C + \mathcal{W}(x)), \quad (5)$$

where $\mathcal{W}(x)$ is the word embedding of input sequence $x$, and $\texttt{enc}_f$ is encoder module of the *forward* model. Before feeding the representation $H_f(x)$ into the decoder, according to the target domain ($\mathbb{S}_0$ or $\mathbb{S}_1$), $H_f(x)$ is concatenated with another style embedding:

$$H_f^\xi(x) = [E_\xi, H_f(x)], \quad (6)$$

where $\xi \in \{C, S_0, S_1\}$. In this way, the hidden representations are shifted according to the target domains. We can obtain $H_f^{S_0}(x)$ for dialogue response generation, $H_f^{S_1}(x)$ for stylized dialogue generation and $H_f^{S_1}(y)$ for text style transfer. Similarly, for the backward model (marked by subscript $_b$), we can obtain the features as follows: $H_b^C(y)$ for inverse dialogue response generation, $H_b^C(\tilde{y})$ for inverse stylized dialogue generation and $H_b^{S_0}(\tilde{y})$ for inverse text style transfer. For ease of reference, we set that all the $H$'s are $d$-dimensional vectors.

## 3.3 Quality discriminators

When analyzing the intermediate results, we find that previous work simply which use iteratively back translation cannot obtain proper context sequences because of the poor data quality. In many scenarios, examples of the synthetic context include:

```
I don't know ...
I'm so happy ...
I'm not sure if I should ...
```

Input $\tilde{\boldsymbol{y}}$ : I believe that Brazilian are very sexy.

⇓ *Backward*

Output $\boldsymbol{x}'$ : I'm so excited for Brazil to come out. $[D_x \to \mathbf{0.45}]$

Input $\boldsymbol{x}$ : Do you want to go back to work right away?

⇓ *Forword*

Output $\tilde{\boldsymbol{y}}'$ : I'm gonna to have a rest. $[D_{\tilde{y}} \to \mathbf{0.13}]$

Figure 2: The example of MPDL model discriminators in context and formal style.

based on which the model cannot generate meaningful response. The inadequate text modeling leads to a peak reduction in model performance. As shown in Figure 2. To evaluate the data quality of the intermediate data (i.e., the pseudo contexts $x$ and responses in style $S_1$), we employ two discriminators $D_x : \mathbb{R}^d \mapsto [0, 1]$ and $D_{\tilde{y}} : \mathbb{R}^d \mapsto [0, 1]$, which are used to classify whether an input text is a real one (e.g., written by people) or a synthetic one (e.g., generated by model).

The discriminators will give a weight to the pseudo sequence generated by model, which can be used to judge the degree to which the model parameters can be updated. $D_x$ and $D_{\tilde{y}}$ are implemented as 3-layer feed-forward networks with LeakyReLU activation function, whose input are $H_f(x)$ and $H_b(\tilde{y})$ obtained according to Eqn.(5). The loss function for discriminators is as follows:

$$
\begin{aligned}
\ell_{\mathrm{dis}}^x &= \log D_x(H_f(x)) + \log(1 - D_x(H_b(y))), \\
\ell_{\mathrm{dis}}^{\tilde{y}} &= \log D_{\tilde{y}}(H_b(\tilde{y})) + \log(1 - D_{\tilde{y}}(H_f(y))), \\
\mathcal{L}_{\mathrm{dis}} &= \mathop{\mathbb{E}}_{(x,y)\sim\mathcal{D}_{\mathrm{dia}};(\tilde{y})\sim\mathcal{D}_{\mathrm{sty}}} - (\ell_{\mathrm{dis}}^x + \ell_{\mathrm{dis}}^{\tilde{y}}).
\end{aligned}
\tag{7}
$$

where $H_b(y)$ is used to generate pseudo context $x'$ for inverse dialogue response generation task, $H_f(y)$ is used to generate pseudo response $\tilde{y}'$ for text style transfer task. Note that at inference time, the discriminator is not used, which means that MPDL will not introduce additional inference cost. The output of the discriminators could be regarded as a confidence score for the sample, which is used further to improve the dual loss in the previous section. Mathematically,

$$
\begin{aligned}
\mathcal{L}_{\mathrm{dual},1}^* &= \mathop{\mathbb{E}}_{\tilde{y}\sim\mathcal{D}_{\mathrm{sty}}} - D_x(H_f(x')) \log P(\tilde{y}|x'; f_{x\tilde{y}}), \\
\mathcal{L}_{\mathrm{dual},2}^* &= \mathop{\mathbb{E}}_{x\sim\mathcal{D}_{\mathrm{dia}}} - D_{\tilde{y}}(H_b(\tilde{y}')) \log P(x|\tilde{y}'; f_{\tilde{y}x}),
\end{aligned}
\tag{8}
$$

In training stage, loss function of MPDL model consists of the standard maximum log likelihood loss and the multi-pass dual learning loss. The mixed learning objective function is defined as follows:

$$
\mathcal{L} = \mathcal{L}_{\mathrm{dia}} + \mathcal{L}_{\mathrm{tra}} + \mathcal{L}_{\mathrm{dual},1}^* + \mathcal{L}_{\mathrm{dual},2}^*.
\tag{9}
$$

# 4 Experiment

## 4.1 Experiment setup

**Datasets.** We train and evaluate our method on two benchmarks, TCFC [30], whose responses are of formal style, and Shakespearean Dialogue Generation Corpus (SDGC)[2], whose responses are of Shakespearean style. TCFC is constructed by crawling text from Twitter. Following [9], we sample $217.2k$ (context, informal response) dialogue pairs as $\mathcal{D}_{\mathrm{dia}}$ and $500k$ formal texts as $\mathcal{D}_{\mathrm{sty}}$ from the TCFC dataset. The test set contains 978 manually-crafted dialogue pairs. For the $\mathcal{D}_{\mathrm{tra}}$, we sample $105.5k$ informal-formal parallel sentences from GYAFC [31]. SDGC is constructed by Twitter, Yahoo Answers and Shakespeare [11]. Shakespeare is a collection of Shakespearean plays with an aligned modern version on supervised style transfer task. For the test set, we manually craft 1,000 dialogue pairs and we will release them to the community. The $\mathcal{D}_{\mathrm{dia}}$ is the same as that for TCFC. We extract $18.4k$ (modern English, Shakespearean English) pairs from [11] as $\mathcal{D}_{\mathrm{tra}}$. All Shakespearean English is used as $\mathcal{D}_{\mathrm{sty}}$. The statistic of datasets is shown in Table 5 of Appendix.

**Baselines.** We compare our proposed model with six baselines, including standard Transformer-based dialogue model, multi-task learning method, language model method, hidden state fusion method, pipeline method and the latest end-to-end method. Specifically, (1) **S2S**: a Transformer-based model trained on $\mathcal{D}_{\mathrm{dia}}$ only, where the weights are initialized by DialoGPT [22]. (2) **MTask**: Luan et al. [32] propose a vanilla multi-task learning model, which is trained with both $\mathcal{D}_{\mathrm{dia}}$ and $\mathcal{D}_{\mathrm{sty}}$. (3) **SLM**: Niu and Bansal [23] propose the fusion model by merging the decoder of a sequence-to-sequence model trained on $\mathcal{D}_{\mathrm{dia}}$ and a language model trained on $\mathcal{D}_{\mathrm{sty}}$. (4) **SFusion**: a multi-task learning model proposed by [24], which builds a structured latent space to bridge the conversation modeling and style transfer through jointly learned with $\mathcal{D}_{\mathrm{dia}}$ and $\mathcal{D}_{\mathrm{sty}}$. (5) **S2S+BT**: the responses with style $S_0$ of pipeline approach is produced by the S2S baseline, and then text style transfer model proposed by [33] is used to get a response with style $S_1$. (6) **SRJT**: a Style Routing approach by employing an

---

[2]Our code and dataset are publicly available at https://github.com/CodebaseLi/MPDL

Table 1: Automatic and manual evaluation results on TCFC dataset. The stylized accuracy of the BERT and SVM classifiers on the holdout test set is 93.98% and 89.57% respectively.

| Model | Automatic Metrics | | | | | Manual Metrics | | |
|---|---|---|---|---|---|---|---|---|
| | **BLEU-1** | **BLEU-2** | **Distinct** | **BERT** | **SVM** | **Fluency** | **Relevance** | **Style-Con.** |
| The target style is formal response (i.e., style $S_1$) | | | | | | | | |
| MTask | 6.35 | 0.50 | 29.3 | 37.3 | 50.1 | 0.78 | 0.33 | 0.58 |
| SLM | 12.6 | 0.99 | 42.5 | 85.6 | 87.2 | 0.83 | 0.45 | 0.87 |
| SFusion | 5.51 | 0.28 | **61.0** | 21.9 | 39.0 | 0.77 | 0.32 | 0.57 |
| S2S+BT | 12.1 | 1.25 | 42.0 | 86.3 | 86.8 | 0.79 | 0.31 | 0.65 |
| SRJT | 15.1 | 1.71 | 43.4 | 97.3 | 96.1 | 0.85 | 0.55 | 0.89 |
| MPDL | **16.5** | **2.07** | 51.3 | **98.6** | **97.1** | **0.88** | **0.64** | **0.91** |
| Human | - | - | 62.7 | 89.6 | 85.8 | 0.88 | 0.65 | 0.90 |
| The target style is informal response (i.e., style $S_0$) | | | | | | | | |
| S2S | 6.92 | 0.61 | 54.8 | 70.1 | 60.9 | 0.75 | 0.46 | 0.66 |
| SFusion | 4.61 | 0.22 | **62.8** | **70.3** | **61.1** | 0.66 | 0.34 | **0.78** |
| SRJT | 6.96 | 0.67 | 49.4 | 69.4 | 59.2 | 0.81 | **0.57** | 0.73 |
| MPDL | **7.12** | **0.69** | 49.5 | **70.3** | 60.7 | **0.83** | 0.55 | 0.75 |
| Human | - | - | 72.6 | 72.0 | 72.1 | 0.79 | 0.56 | 0.78 |

inverse dialogue model with a Joint Training process proposed by [9], in which it is based on back translation. More baselines and evaluating metrics [34] are available in Appendix.

**Training details.** The DialoGPT (small-size 117M) [22] with 12 layers and 768 hidden sizes is used to initialize the multi-pass dual learning modules, which is released by Transformers[3]. Our model has a 50,257 vocabulary size and was trained on Nvidia GTX1080Ti machines with a batch size 10. The maximum input length and maximum output length are set as 45. We choose the Adam optimizer. The learning rate of generators is $2.25 \times 10^{-4}$ with warm-up steps $1 \times 10^3$, while that for discriminators is $3 \times 10^{-4}$. We use the grid search to tune the hyper-parameters.

## 4.2 Main results

**Automatic evaluation.** In this work, we evaluate the generation quality of both style $S_1$ and style $S_0$, following the evaluation metrics in [9]. Content relevance, diversity and style consistency are our primary consideration. For the content relevance, we choose the BLEU score [35] to between the generated response and the ground-truth one. We focus on the unigram precision (BLEU-1) and bi-gram precision (BLEU-2). For diversity, we choose the distinct score [36], which is the ratio between the number of unique $n$-grams and the number of all $n$-grams in the generated responses. In this work, $n$ is 2. For style consistency, we use two text style classifiers, a BERT-based model [37] and an SVM, to evaluate the style consistency, both of which are trained on the GYAFC and Shakespeare datasets. The accuracy of the BERT and SVM classifiers on the holdout test set is 93.98% and 89.57% respectively for the TCFC experiments, and 95.79% and 82.28% respectively for the SDGC dataset. Each setting is independently run five times and we report the mean value of each metric.

The automatic results are summarized in the left part of Table 1 (for TCFC) and Table 2 (for SDGC). In each table, the generation quality of both styles $S_1$ and $S_0$ are evaluated. Overall, our method achieves the highest BLEU-1 and BLEU-2 scores on the two tasks, which shows the superiority of our method. For formal response generation task on the TCFC dataset, compared with the previous state-of-the-art algorithm SRJT, MPDL improves the BLEU-1, BLEU-2 and distinct scores by 1.4, 0.36 and 7.9. The reasons are two-fold: (1) multi-pass dual learning framework significantly improves the learning ability since it bridges the relation between three domains; (2) the discriminator controls the quality of pseudo data. SFusion gets a better distinct score than MPDL, but has worse BLEU-1 and BLEU-2 scores. This is because the discriminator tends to maintain the consistency of the generated responses and the target style, at the cost of reducing diversity. The BERT and SVM scores of our method are also notable, which indicates that our model can generate target style response

---
[3]https://github.com/huggingface/transformers

Table 2: Automatic and manual evaluation results on SDGC dataset. The stylized accuracy of the BERT and SVM classifiers on the holdout test set is 95.79% and 82.28% respectively.

| Model | Automatic Metrics | | | | | Manual Metrics | | |
|-------|--------|--------|----------|------|-----|---------|-----------|------------|
| | BLEU-1 | BLEU-2 | Distinct | BERT | SVM | Fluency | Relevance | Style-Con. |
| The target style is Shakespearean response (i.e., style $S_1$) | | | | | | | | |
| MTask | 5.64 | 0.15 | 30.2 | 18.4 | 20.7 | 0.45 | 0.23 | 0.19 |
| SFusion | 6.76 | 0.21 | 41.6 | 20.1 | 16.0 | 0.50 | 0.27 | 0.21 |
| S2S+BT | 9.68 | 0.38 | 37.4 | **84.7** | **73.0** | 0.64 | 0.33 | 0.70 |
| SRJT | 12.2 | 0.89 | 43.1 | 65.0 | 53.4 | 0.67 | 0.59 | 0.71 |
| MPDL | **13.9** | **1.43** | **53.2** | 64.3 | 54.1 | **0.79** | **0.71** | **0.80** |
| Human | - | - | 81.2 | 92.9 | 66.0 | 0.84 | 0.75 | 0.83 |
| The target style is modern response (i.e., style $S_0$) | | | | | | | | |
| S2S | 9.33 | 0.90 | 43.2 | 93.1 | 94.3 | 0.64 | 0.31 | 0.55 |
| SFusion | 5.71 | 0.12 | 45.7 | 91.5 | 93.9 | 0.45 | 0.41 | 0.63 |
| SRJT | 8.90 | 0.63 | 40.5 | 96.9 | 95.4 | 0.71 | 0.54 | 0.64 |
| MPDL | **10.4** | **1.19** | **46.7** | **97.8** | **96.7** | **0.73** | **0.57** | **0.70** |
| Human | - | - | 79.9 | 92.9 | 92.0 | 0.80 | 0.69 | 0.77 |

better with the auxiliary text transfer task. On informal response generation, by using MPDL, the performances are also improved, where we also achieved the best BLEU-1 and BLEU-2 scores. We observe that compared to the formal case, the BERT score is similar to previous best result, and the SVM score is lower, which is caused by the lack of discriminator for the informal domain. We will explore it in the future.

For the SDGC dataset, MPDL model also shows great advantages in most metrics. We observed that S2S+BT achieved the highest BERT and SVM scores in Shakespearean style. This shows that the unsupervised pipeline method does help to migrate text to the target style, but at the same time, it will lose the conversation information since the style transfer model does not take the context into consideration. MPDL achieves the second achievement in the style classifier while other metrics are improved significantly. SFusion has a much higher Distinct score for the TCFC dataset than it does for SDGC. Because SFusion learns the patterns from the data of the style $S_1$ domain, and more data is helpful to increase the diversity. TCFC contains $500k$ non-parallel data of style $S_1$ domain while SDGC has only $18.4k$. Therefore, MPDL has a better distinct score on the SDGC dataset. We further investigate the generalization ability of MPDL by evaluating modern responses, and find that the model generalizes well not only over different styles but also over different datasets.

**Manual evaluation.** Complementary to automatic evaluation, we also carry out the manual evaluation. We focus on three aspects: (1) *fluency*, which indicates how smooth the sentence is; (2) *relevance*, which shows the content of the response is consistent with the context; (3) *style consistency*, which evaluates how the response style matches the target domain. For a fair comparison, all generated responses and human responses are re-capitalized, de-tokenized and shuffled order. We sample 100 instances from the text set and employ three graduated annotators to rate each context-response pair. The score range of sample is from 0 to 1, where 1 indicates the best score. We provide the responses (generated by models and ground-truth by humans) for each example to our annotators. Every sample can obtain three human judgments and the final scores are averaged across different judges.

The manual evaluation scores are summarized in the right part of Table 1 and Table 2, where we can see that MPDL significantly outperforms previous baselines. Specifically, for formal response generation on TCFC, the scores of our method are close to the human score. On informal text generation, SFusion and SRJT achieved slightly better results on style consistency and relevance than our method. On SDGC, our method achieves the best human evaluation scores, which in general shows the effectiveness of our method. Compared to SFusion and SRJT in style $S_0$, the relevance and style consistency scores of MPDL decrease on TCFC but increase on SDGC. The reason is that we use more unlabeled target domain data for TCFC than SDGC ($500k$ v.s. $18.4k$). Considering that we share the parameters of all tasks, the model for SDGC is less specialized to the target domain

Table 3: Automatic evaluation average results of ablation models for responses with style $S_0$ and $S_1$ on TCFC dataset.

| Model | BLEU-1 | BLEU-2 | Distinct | BERT | SVM |
|---|---|---|---|---|---|
| Ours | **12.0** | **1.38** | 50.1 | **84.4** | 79.2 |
| w/o Discriminator $D$ | 10.9 | 1.33 | **52.8** | 83.1 | 79.4 |
| w/o Multi-Pass | 9.89 | 0.89 | 45.1 | 82.3 | 75.4 |
| w/o DialoGPT | 10.3 | 0.87 | 33.6 | 84.0 | **79.7** |

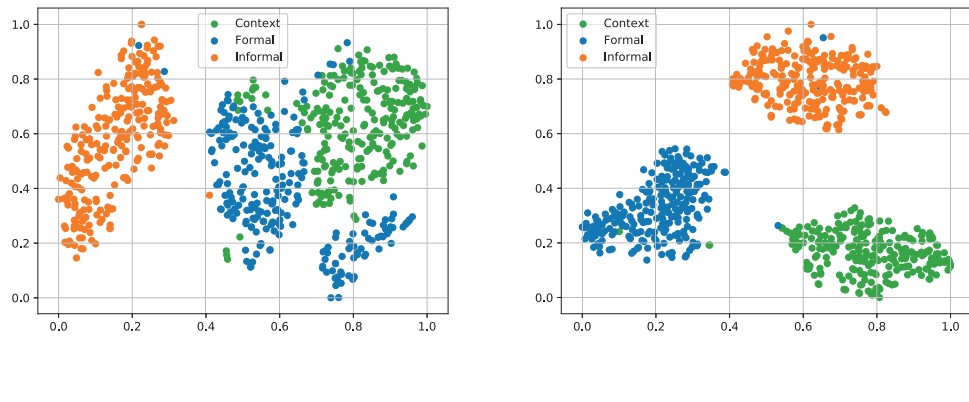

(a) Without discriminators.                   (b) With discriminators.

Figure 3: Visualizing the representations of the context, informal and formal text from TCFC dataset.

and maintains the generalization ability across different domains. More details and the variance are available in the Appendix.

### 4.3 Discussions

**Ablation studies.** To understand the roles of discriminator, multi-pass dual learning (specifically, $\mathcal{L}_{\text{tra}}$ in Eqn.(4)), and pre-trained model in learning to generate stylized responses, we conduct ablation study by removing each component from our framework, and the results are in Table 3. We have the following observations: (1) Without the discriminator, all pseudo responses and context are equally treated. Both the BLEU and BERT scores drop, but the distinct scores increased. This is consistent with our previous explanation, since there is no discriminator to enforce that the styles of generated sequences should match the target domain. (2) Without the $\mathcal{L}_{\text{tra}}$ in Eqn.(4), the results of all metrics significantly drop due to the lack of a closed loop; (3) The pre-training is also helpful to the whole scheme, without which all metrics drop. The ablation study shows the importance of each component in our model. Moreover, we also provide some examples in Table 7 of Appendix to show how the discriminator weights the sequences. Specifically, in dialogue generation of Shakespearean style, the discriminator assigns a score $0.99$ to the sentence "Tis a brave lady", where "Tis" frequently appeared in Shakespearean play. In contrast, the response "I'm not sure what to do with it", which is quite general, obtains a $0.14$ weight.

**Latent space visualization.** To further study the effect of the discriminators, we visualize the hidden state on TCFC dataset, including the context, informal responses and formal responses. We use the output of the last layer in decoder as the representation of the sequences and utilize t-SNE algorithm [38] for visualization. We compare the models trained without the discriminator and with the discriminator. The results are in Figure 3, the remaining visualization results on SDGC dataset and more details are included in the Appendix. As shown in Figure 3(b), after using discriminators, the three types of sequences are properly separated. Without the discriminators (see Figure 3(a)), we can see that the boundary of context and the informal text is not quite clear, which verifies that the quality the synthetic data is low, and we need a module to re-weight them.

Table 4: The example responses produced by MPDL model and the baselines on TCFC dataset.

| Context | A wedding ring? But why would you throw it away? |
|---|---|
| The target style is formal response (i.e., style $S_1$) | |
| Human | Are you sure? You must have surely at least pawned it. |
| MTask | I wish I had a chance to see that. |
| SFusion | They already let me set up! :) |
| SRJT | Yes, they should have given it to you. It is a wedding ring. |
| Ours | I have never thrown away a wedding ring before. |
| The target style is informal response (i.e., style $S_0$) | |
| Human | Right? Surely you would at least have pawned it. |
| S2S | I don't want to talk about it. |
| SFusion | I've emailed the proof when picks it in. I'll send it shortly. |
| SRJT | I'm sure you didn't throw it. |
| Ours | You know you wouldn't throw the engagement ring away. |

**Parallel data size.** To show the effect of the parallel conversation data $\mathcal{D}_{\text{dia}}$ size, we re-train MPDL, MPDL w/o discriminator and the SRJT baseline on TCFC dataset. The BLEU-1 is shown in Figure 4. The performances of all three methods increase w.r.t. to the labeled data size. In comparison, MPDL is less sensitive to the amount of labeled data compared to SRJT, and achieves the best results in BLEU-1, BLEU-2 and Distinct. Moreover, the MPDL with $50k$ labeled dialogue conversation data achieves comparable (even slightly better) results to the SRJT with $200k$ labeled data, demonstrating that our method can effectively utilize data. Experiment of parallel style transfer data $D_{\text{tra}}$ with different sizes are provided in the Appendix.

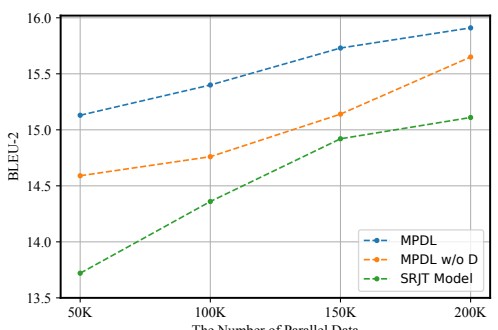

Figure 4: The impact of parallel conversation data quantity on model performance of BLEU-1.

**Case study.** We show some examples of the stylized response in Table 4, and make a comparison of the previous algorithms. For formal English generation (no parallel data between context and the formal response), the MTask model, whose weights are not initialized from the pre-trained model, tends to generate general and bland responses because of the lack of prior knowledge in the dialogue model. SFusion generates the wrong style, where there is an emoji ":)" at the end, since there is no discriminator. For SRJT, although it generates a formal sentence, the response is not consistent to the context, because we would like to see the response to "why would throw it away". Overall, our model performance is the best, capturing more input information and the target style. For informal English generation where parallel data is available, by using our training scheme, we also observe some improvement over the baselines. MPDL model reduces the use of words like "I don't ..." or "I'm sure ..." and tends to generate more meaningful words to make the response more interesting.

## 5 Conclusions and future work

In this paper, we propose a novel stylized dialogue generation model with multi-pass dual learning (MPDL), that can effectively leverage unlabeled data. Compared with standard dual learning, we employ a discriminator to ensure that the responses move towards the target style while the content is relevant. Experiments on two stylized response generation benchmark datasets demonstrate that our method has significant advantages over the recent works, and our model achieves state-of-the-art results in the formality and Shakespearean style response generation. For future work, first we can extend the stylized dialogue from single style to multiple styles without too much additional cost; second, we will increase the robustness of our model; third, we will further explore how to effectively use more unlabeled data.

## Broader Impact

Generating stylized responses is an important step towards real artificial intelligence. It can be predicted that its achievements can significantly improve the experience of human-computer interaction, and can be applied in various fields, such as education, finance, and so on. This work has the potential positive impact on an intelligent and engaging dialogue system. At the same time, this work may have some negative consequences in social interaction. Besides, if the pre-training model is covered by malicious and vulgar information, it will bring bad information feedback to users. Therefore, we should be cautious of these advantages and disadvantages.

## Acknowledgments

This work was supported by the National Key Research and Development Program of China (No. 2020AAA0106600) and NSFC Grant (No. 62122089 and No. 61876196).

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
