# Appendix of Stylized Dialogue Generation with Multi-Pass Dual Learning

**Jinpeng Li[1], Yingce Xia[2], Rui Yan[3,5*], Hongda Sun[3], Dongyan Zhao[1,4*], Tie-Yan Liu[2]**
[1]Wangxuan Institute of Computer Technology, Peking University, Beijing, China
[2]Microsoft Research Asia, Beijing, China
[3]Gaoling School of Artificial Intelligence, Renmin University of China, Beijing, China
[4]Institute for Artificial Intelligence, Peking University, Beijing, China
[5]Beijing Academy of Artificial Intelligence, Beijing, China
lijinpeng@stu.pku.edu.cn, {yingce.xia, tyliu}@microsoft.com,
{ruiyan, sunhongda98}@ruc.edu.cn, zhaody@pku.edu.cn

## A   Appendix

### A.1   Algorithm

The MPDL model is trained with the following Algorithm 1:

---
**Algorithm 1** The training process

---
**Require:** Parallel data $\mathcal{D}_{\text{dia}}$, $\mathcal{D}_{\text{tra}}$, unpair data $\mathcal{D}_{\text{sty}}$ and parameters $\overrightarrow{\Theta}, \overleftarrow{\Theta}$
**Ensure:** The MPDL model $f_{x\tilde{y}}$

1: Initialize the forward and backward encoder-decoder parameters $\overrightarrow{\Theta}, \overleftarrow{\Theta}$ using DialoGPT
2: Define $N \leftarrow$ freeze steps
3: **repeat**
4:     Sample $b$ mini-batch size dialogue pairs $\mathcal{D}_{\text{dia}}^b \subset \mathcal{D}_{\text{dia}}$
5:     Sample $b$ mini-batch size transfer pairs $\mathcal{D}_{\text{tra}}^b \subset \mathcal{D}_{\text{tra}}$
6:     Train $f_{xy}$, $f_{yx}$ by obtaining Loss $\mathcal{L}_{\text{dia}}$ on $\mathcal{D}_{\text{dia}}^b$
7:     Train $f_{y\tilde{y}}$, $f_{\tilde{y}y}$ by obtaining Loss $\mathcal{L}_{\text{tra}}$ on $\mathcal{D}_{\text{tra}}^b$
8:     **if** Current Step $> N$ **then**
9:         Sample $b$ mini-batch size style sentences $\mathcal{D}_{\text{sty}}^b \subset \mathcal{D}_{\text{sty}}$ for backward stage
10:         Sample $b$ mini-batch size contexts $\mathcal{D}_{\text{dia}}^b \subset \mathcal{D}_{\text{dia}}$ for forward stage
11:         Decode the pseudo context $x'$ and $\tilde{y}'$ from $f_{yx}(f_{\tilde{y}y}(\tilde{y}))$ and $f_{y\tilde{y}}(f_{xy}(x))$
12:         Train $f_{x\tilde{y}}$, $f_{\tilde{y}x}$ by obtaining Loss $\mathcal{L}_{\text{dual},1}^*$, $\mathcal{L}_{\text{dual},2}^*$ on $\{x', \tilde{y}\}$, $\{x, \tilde{y}'\}$
13:     **end if**
14:     Optimizing MPDL model with the mixed Loss $\mathcal{L}$
15: **until** The model converge
16: **return** $f_{x\tilde{y}}$

---

### A.2   Datasets

Table 5 shows the statistics of datasets, including the number of data and the average length of sentences. By analyzing the corpus statistics in terms of the average length, we can assume that the text transfer datasets are in the same content space for the dialogue datasets. For the test set of SDGC, we manually craft 1,000 dialogue pairs and we will release them to the community. Specifically, we

---

*corresponding authors: Rui Yan (ruiyan@ruc.edu.cn) and Dongyan Zhao (zhaody@pku.edu.cn).

35th Conference on Neural Information Processing Systems (NeurIPS 2021).

Table 5: The statistics of datasets.

| Datasets | | Train | Test |
|---|---|---|---|
| **TCFC** | $\mathcal{D}_{\mathrm{dia}}$ | 217,222 | 978 |
| | $\mathcal{D}_{\mathrm{sty}}$ | 500,000 | |
| | Avg. | 12.44 | 14.49 |
| **GYAFC** | $\mathcal{D}_{\mathrm{tra}}$ | 104,562 | 2,748 |
| | Avg. | 10.69 | 10.71 |
| **SDGC** | $\mathcal{D}_{\mathrm{dia}}$ | 217,222 | 1000 |
| | $\mathcal{D}_{\mathrm{sty}}$ | 18,395 | |
| | Avg. | 12.07 | 9.12 |
| **Shakespeare** | $\mathcal{D}_{\mathrm{tra}}$ | 18,395 | 1,462 |
| | Avg. | 11.38 | 7.77 |

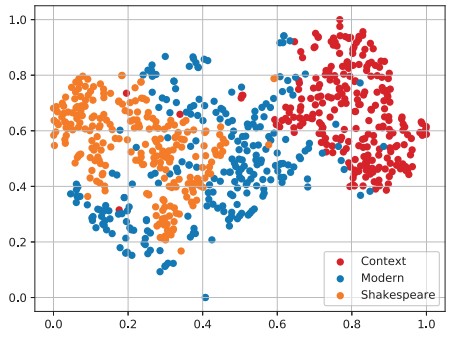

(a) Without discriminators.

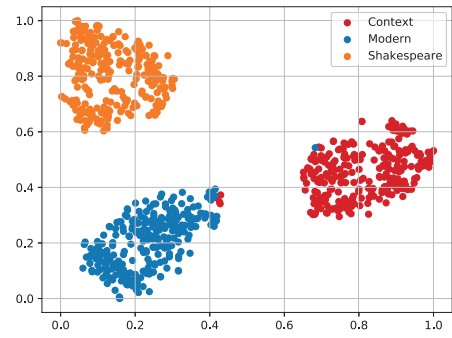

(b) With discriminators.

Figure 5: Visualizing the representations of the context, modern and Shakespearean text from SDGC.

train a modern response-to-context model with the conversation dialogue dataset at first. The pseudo contexts can be generated by feeding the above model with the test set of Shakespeare. Generally, the relevance, information, and diversity of the pseudo contexts are low-level, thus the real value of the data is discounted. Therefore, we train different models through various model structures and iterations, and generate six contexts for each response. The final and most important step is manually crafting the texts based on crowdsourcing. We hired graduate students as annotators to rewrite a new context according to the six pseudo contexts and original responses. We annotated 50 samples randomly extracted from the dataset, and calculated our average annotation time so we could set a fair salary for annotators' training annotation. During the training annotation process, they were paid as well. We also calculated the average annotation time for each dialogue during training, based on which we determined the final salary was around 9.4 dollars per hour. This hourly salary was the same for manual checking.

## A.3 More Results on Automatic and Manual Evaluation

Figure 5 shows the hidden state of the last layer of the model decoder after dimension reduction using PCA on the SDGC dataset. Figure 6 shows the influence of the parallel data $\mathcal{D}_{\mathrm{dia}}$ size of the BLEU-2 and Distinct on the TCFC dataset.

Table 6 provides the effect of different complexity discriminators on the performance of formal response on TCFC dataset. We find that it is not the complex discriminator that achieves better results. 8-layer Transformer encoder gets a higher distinct but a lower BLEU. In consideration, MPDL adopts a three-layer 3-layer feed-forward network as discriminator.

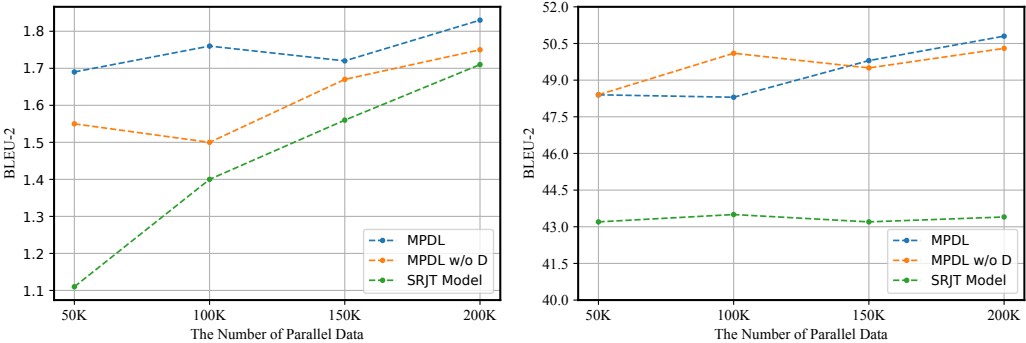

(a) The performance of BLEU-2.      (b) The performance of Distinct.

Figure 6: The impact of parallel conversation data quantity on model performance.

Table 6: The performance of different discriminators in formal response on TCFC.

| Discriminators | BLEU-1 | BLEU-2 | Distinct | BERT | SVM |
|---|---|---|---|---|---|
| 3-layer feed-forward network | **16.5** | **2.07** | 51.3 | **98.6** | **97.1** |
| 2-layer Transformer encoder | 15.0 | 1.81 | 50.4 | 98.4 | 96.4 |
| 8-layer Transformer encoder | 14.4 | 1.62 | **52.8** | 95.6 | 93.7 |

Table 7: The example of weights produced by discriminators.

| Discriminators | Pseudo Text | Weight |
|---|---|---|
| Context $D_x$ | She's such a great actress and I love her! | 0.99 |
| | What's the best way to get a job in my life? | 0.87 |
| | I don't know what to say. | 0.19 |
| Formality $D_{\tilde{y}}$ | You have a very precious soul. | 0.99 |
| | I will share that pack with you! | 0.68 |
| | I would like to see U. | 0.09 |
| Shakespearean $D_{\tilde{y}}$ | Tis a brave lady. | 0.99 |
| | I'll be here and you're welcome. | 0.21 |
| | I'm not sure what to do with it | 0.14 |

Table 7 shows the weights given by different discriminators for the pseudo sentences in the middle. $D_x$ tends to give more weight to sentences with more notional words. Specifically, "U" is an informal expression of "you", formality $D_{\tilde{y}}$ gives a weight of 0.09. Similarly, "Tis" is topical word in the Shakespearean plays. Shakespearean $D_{\tilde{y}}$ predicts a weight of 0.99.

We compare baseline and many variant models of MPDL on TCFC dataset, the results are in Table 8. The supervised pipelined method, where the first model is to generate the response in style $S_0$ and the second model is to transfer it into the response in style $S_1$ in a supervised manner (Pipeline). The experimental results show that MPDL significantly outperforms the supervised pipelined method. We pre-train the discriminator for 8000 steps, and then keep it fixed when training MPDL (MPDL_PreD). The performance of pretrained discriminators is not as good as joint training. The model (MPDL_Unpair) using non-parallel style transfer $\mathcal{D}_{\text{tra}}$ data performs significantly better than the baselines like Pipeline and S2S+BT. The non-parallel text transfer resources are easy to obtain. We will continue to explore how to accelerate the model and make it more practical. From MPDL w/o $D_x$ and MPDL w/o $D_{\tilde{y}}$, we find that removing any discriminator hurts the performances. Both of them contribute to our task: (1)With $D_{\tilde{y}}$, the generated sequence should match style $S_1$ (i.e., the target style), without which the BERT and SVM scores of "formal style" become lower. (2)With $D_x$, the quality of the pseudo context data $x'$ is evaluated, without which the BLEU-1 and BLEU-2 drop in "formal style" generation.

Table 8: The performance of different models on TCFC.

| Model | BLEU-1 | BLEU-2 | Distinct | BERT | SVM |
|-------|--------|--------|----------|------|-----|
| The target style is formal response (i.e., style $S_1$) | | | | | |
| MPDL | 16.5 | 2.07 | 51.3 | 98.6 | 97.1 |
| MPDL_PreD | 16.2 | 2.01 | 50.4 | 98.2 | 97.1 |
| MPDL_Unpair | 15.2 | 1.76 | 48.5 | 98.7 | 96.9 |
| MPDL w/o $D_x$ | 15.6 | 1.72 | 47.0 | 98.9 | 96.6 |
| MPDL w/o $D_{\tilde{y}}$ | 15.7 | 1.76 | 50.4 | 97.9 | 95.7 |
| Pipeline | 12.9 | 1.33 | 43.1 | 94.5 | 92.7 |
| The target style is informal response (i.e., style $S_0$) | | | | | |
| MPDL | 6.92 | 0.69 | 49.3 | 70.3 | 60.7 |
| MPDL_PreD | 6.60 | 0.58 | 47.2 | 70.7 | 60.2 |
| MPDL w/o $D_x$ | 6.78 | 0.63 | 43.6 | 68.9 | 60.6 |
| MPDL w/o $D_{\tilde{y}}$ | 6.32 | 0.47 | 44.5 | 70.1 | 59.4 |

We conduct experiments on TCFC dataset with different sizes of $D_{\mathrm{tra}}$. The formal responses results are in Table 9. There is a positive correlation between the size of $D_{\mathrm{tra}}$ and BLEU. When MPDL uses 40k $D_{\mathrm{tra}}$, the performance of MPDL has surpassed the baseline SRJT in all indicators.

Table 9: The formal performance of different sizes of $D_{\mathrm{tra}}$ on TCFC.

| Model | BLEU-1 | BLEU-2 | Distinct | BERT | SVM |
|-------|--------|--------|----------|------|-----|
| $\mathrm{SRJT}_{Zero}$ | 15.1 | 1.71 | 43.4 | 97.3 | 96.1 |
| $\mathrm{MPDL}_{ALL(104k)}$ | 16.5 | 2.07 | 51.3 | 98.6 | 97.1 |
| $\mathrm{MPDL}_{80k}$ | 15.7 | 1.80 | 48.2 | 98.7 | 97.9 |
| $\mathrm{MPDL}_{60k}$ | 15.6 | 1.78 | 49.0 | 98.1 | 97.5 |
| $\mathrm{MPDL}_{40k}$ | 15.3 | 1.83 | 50.3 | 98.4 | 97.2 |
| $\mathrm{MPDL}_{20k}$ | 14.8 | 1.38 | 48.6 | 98.5 | 97.7 |

For making results easy to compare between different models, we use the aggregation strategy [1] to combine the three metrics (BLEU-1, BERT and SVM) into a single score J(B,B,S), the results with formal stylized responses in Table 10 prove that our MPDL model still performs well.

Table 10: The formal performance of different models on TCFC.

| Model | BLEU-1 | BERT | SVM(89.5%) | J(B,B,S) |
|-------|--------|------|------------|----------|
| SRJT | 15.1 | 97.3 | 96.1 | 12.6 |
| MPDL | 16.5 | 98.6 | 97.1 | 13.4 |

For each manual indicator, we give the annotators three ranges: $0 \sim 0.33$ for very dissatisfied; $0.34 \sim 0.66$ for general satisfaction; and $0.67 \sim 1$ for very satisfied. We annotated 50 samples randomly extracted from the test set, and calculated our average annotation time so we could set a fair salary for annotators' training annotation. During the training annotation process, they were paid as well. We also calculated the average annotation time for each dialogue during training, based on which we determined the final salary was around 9 dollars per hour. This hourly salary was the same for SDGC dataset. The variance of TCFC dataset in formal responses in Table 11.

Table 12 present some examples from TCFC and SDGC respectively. In each case, we show the dialogue context, the human response and responses from different models (SRJT and MPDL). We can see that responses from MPDL are doing well, particularly 'oft' is a typical Shakespearean word. This shows that MPDL can learn good knowledge transfer ability with the help of a small amount of style data.

Table 11: The variance of TCFC dataset in formal responses.

| Model | Fluency | Relevance | Style-Con. |
|---|---|---|---|
| SFusion | 0.027 | 0.017 | 0.020 |
| SRJT | 0.017 | 0.012 | 0.016 |
| MPDL | 0.015 | 0.011 | 0.015 |
| Human | 0.006 | 0.005 | 0.013 |

Table 12: The example responses produced by MPDL model and the baselines on TCFC and SDGC datasets.

| Context | I'm not being nominated pabo noona. |
|---|---|
| The target style is formal response (i.e., style $S_1$) | |
| Human | Are you serious? It is funny but I saw a tweet saying just that. Sorry! |
| SRJT | Yes, I can agree with you, but I am not even the one. |
| Ours | It is obvious that pabo is the only one who needs to be nominated for this award. |
| The target style is informal response (i.e., style $S_0$) | |
| Human | really? hahahaha but i saw some tweet saying that -_- sorry! |
| SRJT | i'm not even an actor. |
| Ours | lol you're not even the first person I have told you. |
| **Context** | You scolded me often for loving Rosaline. |
| The target style is Shakespearean response (i.e., style $S_1$) | |
| Human | For doting, not for loving, pupil mine. |
| SRJT | You can't even get my name wrong. |
| Ours | But I prated you oft to love Rosaline. |
| The target style is modern response (i.e., style $S_0$) | |
| Human | I scolded you for obsessing about her, not for loving her, my student. |
| SRJT | haha I didn't. I'm still confused. |
| Ours | I did it for a reason. |

## A.4 Computing Infrastructure

We mainly use Python3.6[2] as our coding language. Key libraries used in our code include Pytorch 1.4.0[3], Transformers 4.0.1[4] and the others that we specify the requirement list when we formally release our code. And we run our experiments on a single Nvidia GTX 1080 Ti GPU machine for 1.5 days.