# OpenReview forum: "Stylized Dialogue Generation with Multi-Pass Dual Learning"
_NeurIPS.cc/2021/Conference — NeurIPS 2021 Poster_

### Official Review · Reviewer_tHTV · 2021-07-05

**Rating:** 6
**Confidence:** 4

**Summary:**

This paper studies the task of stylized dialogue generation, where the goal is to generate dialogue responses in a particular target style (formal / informal / Shakespeare-like). The authors operate in a setting where they do not have access to any dialogue training data in the target style, but instead have access to dialogue data in a complementary style, along with parallel data to transfer between the two styles. For instance, the authors study dialogue generation in formal style, and assume access to dialogue data in informal style, parallel data between formal / informal sentences, and a corpus of unpaired formal sentences.

The authors model this task as a three-way dual learning problem, a technique which has proven useful for machine translation [1]. They jointly training their model to perform 1) style transfer in both directions; 2) dialogue / inverse-dialogue generation to the complementary style; 3) dialogue / inverse-dialogue generation to the target style. 1 / 2 act as auxiliary tasks to aid (3) via shared models. Since no parallel data is available for (3), the authors perform sampling (using current version of model for 1 / 2) to create pseudo parallel data for (3). Since sampling can be noisy, the authors additionally use discriminators to weight samples, which judge whether the samples are real/fake (akin to adversarial training).

The authors evaluate their approach on formal dialogue generation and Shakespearan dialogue generation. They compare their approach to 5-6 baselines, along with human upperbounds. Both automatic and human evaluation is conducted. The authors additionally conduct ablation studies, analyze their discriminator, and show the effect of the size of parallel data.

[1] - https://arxiv.org/abs/1611.00179

**Limitations And Societal Impact:**

Yes

**Main Review:**

**Strengths**

1. Style-controlled text generation is an important problem in natural language generation technology, and is far from solved. This has several important applications in chatbots and coversation agents.

2. The authors use several interesting ideas in this paper, which were new to me. The idea to use dual learning to boost style-controlled generation as well as discriminators for weighing poor synthetic data were interesting and could be more widely applicable in style transfer literature.

3. The authors approach beats several strong baselines on automatic & human evaluation. Several good analysis experiments have been conducted in the paper including ablations, analysis of the effect of discriminators and the effect of the size of available parallel data.

**Weaknesses**

1. My biggest issue with the paper is the usage of parallel data. Most style transfer literature [1, 2, 3, 4] operates in an unsupervised / unpaired setting (with access to D_sty only), and recent work [5] is assuming access to only 5-10 exemplars from D_sty. Access to parallel data is a strong assumption --- besides formality, shakespeare, sentiment transfer, almost no parallel resources exist. Moreover, all these datasets are in English, and no parallel style transfer resources exist for other languages. Even assuming access to dialogue data in a single style (`D_dia`) is strong assumption in my opinion. Indeed, the authors show that their method's performance reduces significantly if you don't have access to this parallel data (Table 3 w/o Multi-Pass, Figure 4), which reduces the applicability of this work.

2. The evaluation in this paper could be improved. Here are some specific thoughts ---

* Since there are three orthogonal metrics [7] for both human / automatic evaluation (style accuracy, semantic correctness, fluency), it's hard to compare between systems when only one metric improves (and the other does not), for instance SFusion vs MPDL in Table 1; S2S+BT vs MPDL in Table 2; experiments in Table 3. I suggest using the aggregation strategy from [2] to combine the metrics into a single score, making it easy to compare between systems.

* The BLEU scores are quite low overall for all systems --- which is not surprising given the large space of possible outputs which makes evaluation challenging and easy to game [8]. I suggest using other automatic metrics too, such as those based on natural language inference [9] or the recent BEGIN benchmark [10].

* Since automatic metrics can be flawed, human evaluation is critical. While the authors conduct human evaluation, more details will be helpful. Specifically, i) Were the annotators familiar with the research goals? (ideally they should be unfamiliar to avoid bias) ii) Were they shown outputs from different systems in a blind manner (not aware which system produced which output) & shuffled order? (consecutive samples from different systems) iii) What's the agreement between annotators & variance / error-bars of their annotation? Are the gains statistically significant? iv) How much were the annotators paid for this experiment?

* More qualitative outputs from the system would be very helpful. Currently only two outputs are shown in Table 4. In both examples it's not really clear to me if the outputs are really formal / informal, which makes me doubt the quantiative evaluation in this work (In second example, "wouldn't" has an apostrophe which shouldn't be there in an informal style generation).

**Minor**: The notation in Section 3 is confusing, primarily due to lots of inverted commas and tildes. I suggest avoiding these and using subscripts with short natural language identifiers (like you did for `D_dia / D_tra / D_sty`). Also it will be good to mention `BERT / SVM` in Table 1 refer to classification accuracy (my initially incorrect thought reading this was a reference to BERTScore [6], which confused me).

**Overall**

In terms of the NeurIPS reviewer guidelines, I think the paper has decent originality (strength #2), average quality (strength #3, weakness #2), average clarity (minor #1), and average significance (strength #1, weakness #1). Overall, I liked the ideas in the paper, but I think weakness #1 is the biggest deal-breaker for me which will prevent me from giving a higher score. I encourage the authors to improve this in the next version of the paper.

[1] - https://arxiv.org/abs/2011.00416
[2] - https://arxiv.org/abs/2010.05700
[3] - https://arxiv.org/abs/1804.09000
[4] - https://arxiv.org/abs/1811.00552
[5] - https://arxiv.org/abs/2010.03802
[6] - https://arxiv.org/abs/1904.09675
[7] - https://arxiv.org/abs/1910.03747
[8] - https://arxiv.org/abs/2103.06332
[9] - https://arxiv.org/abs/1904.03371
[10] - https://arxiv.org/abs/2105.00071

-------

**After author response**: Thanks for the excellent response. I've raised my score to 6.

**Time Spent Reviewing:**

4-5 hours

---

> ### Comment · Reviewer_tHTV · 2021-08-10
> **Quick clarification about weakness #1 regarding parallel data**
>
> Hi everyone,
> I just wanted to add a quick clarification about my weakness #1, since I noticed I was unclear in my review. By "parallel data", I'm referring to `D_tra`, or parallel sentences which share semantics but differ in style. This "parallel data" is different from the "parallel data" referred to by the authors in their abstract / TL;DR, where they are referring to parallel dialogue utterances differing in style.

---

> ### Author Response · Authors · 2021-08-10
> **Response to Reviewer tHTV**
>
> Thank you for the valuable comments that help us improve the work. Below we address the concerns mentioned in the review:
>
> *Q1: My biggest issue with the paper is the usage of parallel data. Most style transfer literature [1, 2, 3, 4] operates in an unsupervised / unpaired setting (with access to D_sty only), and recent work [5] is assuming access to only 5-10 exemplars from D_sty. Access to parallel data is a strong assumption --- besides formality, shakespeare, sentiment transfer, almost no parallel resources exist. Moreover, all these datasets are in English, and no parallel style transfer resources exist for other languages.*
>
> A1:  Thanks for your comments.
>
> **1. Parallel text style transfer data make the model converge faster and more stable**
>
> Compared to previous work about general text style transfer, in this work, we focus on stylized dialogue generation, where a response should be of a specific style and be consistent with the context. Simply using a dialogue generator with a style transfer module will not lead to good results, which is the "pipeline" in the table below. Most failure cases of "pipeline" are due to inconsistency between the context and response. Therefore, these are two related but different tasks. We would like to further explore along this direction and in our submission, we choose the setting that both (context, Style S0) and (Style S0, Style S1) have parallel data. The parallel data make the model optimize towards the target hidden space faster.
>
> **2. Preliminary exploration using non-parallel data**
>
> Following your suggestion, we also conduct a preliminary version of MPDL without parallel data between style S0 response and style S1 response. Denote the data in style S0 and style S1 as $U_0$ and $U_1$, which is non-parallel. Following the techniques from unsupervised NMT, the style transfer model $f_{y\tilde{y}}$ and $f_{\tilde{y}y}$ are implemented as follows:
>
> $\max_{f_{y\tilde{y}}}\frac{1}{|U_1|}\sum_{\tilde{y}\in U_1}\log P(\tilde{y}|\texttt{SG}(f_{\tilde{y}y}(\tilde{y})); f_{y\tilde{y}})$
>
> $\max_{f_{\tilde{y}y}}\frac{1}{|U_0|}\sum_{y\in U_0}\log P(y|\texttt{SG}(f_{y\tilde{y}}(y)); f_{\tilde{y}y})$
>
> where SG(…) means "stop gradient" during propagation. A discriminator for style S0 is also used. Denote this method as MPDL_Unpair. The results are shown as follows:
>
> | Model |BLEU-1 | BLEU-2 | Distinct | BERT | SVM |
> |:----:|:----:|:----:|:----:|:----:|:----:|
> | MPDL | 16.5 | 2.07 | 51.3 | 98.6 | 97.1 |
> | MPDL_Unpair| 15.2 | 1.76 | 48.5 | 98.7 | 96.9 |
> | Pipeline | 12.9 | 1.33 | 43.1 | 94.5 | 92.7 |
> | S2S+BT | 12.1 | 1.25 | 42.0 | 86.3 | 86.8 |
>
> We can see that under our proposed framework (i.e., MPDL), even if no parallel data is available between the style S0 and S1, it still outperforms previous baselines like S2S+BT.
>
> **3. Preliminary exploration using a small amount of parallel data**
>
> In addition, we found that using a small amount of parallel data D_tra can achieve satisfactory results, which reduces the difficulty of data acquisition. The experimental results can refer to in the QA3 of Responses to Reviewer FhxC. Therefore, we present the MPDL model in the paper considering the performance.
>
> In general, exploring the unsupervised and supervised setting of text transfer is meaningful work. Thanks for your suggestions and references, and we will definitely follow this direction in the future.
>
> &nbsp;
>
> *Q2: Even assuming access to dialogue data in a single style (D_dia) is strong assumption in my opinion.*
>
> A2: In DIALOGPT [22], the authors collected 147,116,725 dialogue instances from Reddit, which has 1.8 billion words in total. Therefore, this data is not difficult to obtain and this is not a strong assumption.
>
> &nbsp;
>
> *Q3: Indeed, the authors show that their method's performance reduces significantly if you don't have access to this parallel data (Table 3 w/o Multi-Pass, Figure 4), which reduces the applicability of this work.*
>
> A3: In Figure 4, we explore the amount of parallel data between context and response of style 0, not the parallel data between style 0 and style 1. We can usually collect enough data between context and response (not stylized) domains. Compared to previous methods of stylized dialogue translation, with less training data, our method still performs the best.
>
> &nbsp;
>
> *Q4: I suggest using the aggregation strategy from [2] to combine the metrics into a single score, making it easy to compare between systems. The BLEU scores are quite low overall for all systems --- which is not surprising given the large space of possible outputs which makes evaluation challenging and easy to game [8]. I suggest using other automatic metrics too, such as those based on natural language inference [9] or the recent BEGIN benchmark [10].*
>
> A4: Thanks for pointing it out. We use the aggregation strategy from [2] to combine the metrics into a single score J(B,B,S), the results with formal stylized responses in TCFC as follows:
>
> | Model |BLEU-1 | BERT | SVM(89.5%) | J(B,B,S) |
> |:----:|:----:|:----:|:----:|:----:|
> | SRJT | 15.1 | 97.3 | 96.1 | 12.6 |
> | MPDL | 16.5 | 98.6 | 97.1 | 13.4 |
>
> Our method still outperforms SRJT. We will discuss more evaluation methods in the revision according to your comments, such as [9] and [10]. Thank you for the valuable comments that help us improve the work.
>
> &nbsp;
>
> *Q5: Since automatic metrics can be flawed, human evaluation is critical. While the authors conduct human evaluation, more details will be helpful. Specifically, i) Were the annotators familiar with the research goals? (ideally they should be unfamiliar to avoid bias) ii) Were they shown outputs from different systems in a blind manner (not aware which system produced which output) & shuffled order? (consecutive samples from different systems) iii) What's the agreement between annotators & variance / error-bars of their annotation? Are the gains statistically significant? iv) How much were the annotators paid for this experiment?*
>
> A5: i) We hired three graduate students as annotators to evaluate the 100 instances sampled from the text set in two styles. One of them knows our goal and the other two not.
>
> ii) Yes, the outputs are shown in a blind manner and the annotators do not know which system a sequence comes from.  Specifically, all generated responses and human responses are re-capitalized, de-tokenized and the orders are shuffled (Line 244).
>
> iii)  For each metric, we give the variance of TCFC dataset in formal responses as follows:
>
> | Model | Fluency | Relevance | Style-Con. |
> |:----:|:----:|:----:|:----:|
> | SFusion| 0.027 | 0.017 | 0.020 |
> | SRJT | 0.017 | 0.012 | 0.016 |
> |MPDL | 0.015 | 0.011 | 0.015 |
> |Human | 0.006 | 0.005 | 0.013 |
>
> The variance is within the acceptable range, and the gains of our method are statistically significant.
>
> iv) The salary is 9 dollars per hour, and the salary is also paid during the training annotation process. To give a fair salary, we first annotate 50 samples by ourselves, calculate the time and effort and set this amount.
>
> Others: A total of 200 data pairs (100 context-formal and 100 context-informal) were annotated per model, with three annotators per pair. For each indicator, we will give the annotators three ranges:  0\~0.33 for very dissatisfied; 0.34\~0.66 for general satisfaction; and 0.67\~1 for very satisfied.
>
> &nbsp;
>
> *Q6: More qualitative outputs from the system would be very helpful. Currently only two outputs are shown in Table 4. In both examples it's not really clear to me if the outputs are really formal / informal, which makes me doubt the quantiative evaluation in this work (In second example, "wouldn't" has an apostrophe which shouldn't be there in an informal style generation).*
>
> A6: We will show more quantitative analysis in the revision. In the second example, "wouldn't" has an apostrophe which should be there in an informal style generation.
>
> &nbsp;
>
> *Q7: The notation in Section 3 is confusing, primarily due to lots of inverted commas and tildes. I suggest avoiding these and using subscripts with short natural language identifiers (like you did for D_dia / D_tra / D_sty). Also, it will be good to mention BERT / SVM in Table 1 refer to classification accuracy (my initially incorrect thought reading this was a reference to BERTScore [6], which confused me).*
>
> A7: We will carefully revise the symbols in Section 3 and increase the accuracy of Bert and SVM in Table 1.

---

> > ### Comment · Reviewer_tHTV · 2021-08-17
> > **Thanks for the excellent response, I've raised my score to 6**
> >
> > Thanks for the great response and addressing some of my concerns, and I apologize for my mis-interpretation in Q3.
> >
> > Regarding Q2, my concern was more on the lines of, "is it practical to collect a large scale dataset where all responses share the same style? for example, all responses are simplified, or all responses are formal in nature?" But to be fair, such a dataset can be built by running a style classifier on a large-scale dialogue dataset like the one you mentioned.
> >
> > I've raised my score to 6, but I encourage the authors to evaluate the non-parallel `D_tra` setting more thoroughly in the next version of the paper (perhaps using human evaluation).

---

### Official Review · Reviewer_FhxC · 2021-07-15

**Rating:** 7
**Confidence:** 4

**Summary:**

This paper focuses on proposing multi-pass dual learning (MPDL) to build mappings among dialogue context and responses in two styles. The proposed model is reasonable and is the extension of the classical dual learning method, which usually maps between two domains. The idea is reasonable and novel, and the experiments show the improved performance in terms of both automatic and human evaluation for both datasets.

A main concern is about the paired style transfer data $D_{tra}$, which limits the flexibility of usage. Also, an intuitive baseline may be needed for fair comparison. The further suggestion is to remove the "paired style transfer data" constraint, considering that the proposed MPDL may still work.

**Limitations And Societal Impact:**

Yes.

**Main Review:**

This paper proposes multi-pass dual learning (MPDL) to build mappings among three domains (dialogue context $C$, response of style $S_0$, response of style $S_1$) through duality. The target scenario is lack of $(C, S_1)$ parallel data for training, but there is unlabelled data with the style $S_1$.

In the proposed MPDL, the main idea is to utilize the reconstruction property of the response with the style $S_1$, which is mapped to the response with the style $S_0$ via a style transfer model, then mapped to the context via an inverse dialogue generation model, and then finally mapped to the original one (the response with the style $S_0$) via a dialogue generation model. In addition to minimizing the reconstruction error through multiple passes, two discriminators are introduced to ensure the good quality of pseudo contexts and responses in style $S_1$. By jointly learning the mixed learning objective, the proposed method outperforms multiple baselines for two benchmark datasets and most metrics in the conducted experiments.

The idea about using multi-pass dual learning is reasonable and original, where the unlabelled data in style $S_1$ can be effectively utilized and the mappings among three domains can be obtained. However, there are some concerns to be addressed.

- In the experiments, three datasets are used: $D_{dia}$: $(C, S_0)$ paired data, $D_{tra}$: $(S_0, S_1)$ paired data, and $D_{sty}$: unlabelled $S_1$ data. An intuitive baseline can be a pipelined method, where the first model is to generate the response in style $S_0$ and the second model is to transfer it into the response in style $S_1$ in a supervised manner. However, this baseline is not compared in the experiments. Although the unlabelled data $D_{sty}$ is not used, it still needs to be shown in the table so that we can see how much improvement the unlabeled data can bring in. (The S2S+BT baseline uses "unsupervised" text style transfer model [33] instead of "supervised" learning, where the performance is expected worse than the supervised one.)

- In the proposed MPDL framework, it may be possible to use the unpaired $D_{tra}$ for learning $f_{\tilde{y}y}$ and $f_{y\tilde{y}}$ and add a discriminator to check the quality of $y$. Hence, the paired style transfer data may not be required, making the target scenarios more practical. Have the authors tried this setting? (Comparing the results of this setting with S2S+BT is fair.)

- In the discussions, the authors analyze the sensitivity of the size of the parallel data $D_{dial}$. The results are good, but it is also useful to show the impact of the size of $D_{tra}$. Because the main idea is to better utilize unlabelled $S_1$, so the paired style transfer data $D_{tra}$ may not be available/enough. Showing the results with different sizes of $D_{tra}$ may be more informative from my perspective.

- Are the numbers shown in Table 3 for the TCFC dev set? Because the numbers are quite different from ones in Table 1, can they be compared directly?

- For ablation study, removing discriminators shows the degraded performance in Table 3. Can the authors show the results of removing only one discriminator (either $D_x$ or $D_\tilde{y}$), so that we can see which one contributes more?

- The diversity (Distinct) is sometimes lower in the proposed method. Is it because $f_{yx}$ and $f_{x\tilde{y}}$ are not 1-1 mappings and the reconstruction $\tilde{y}'=f_{x\tilde{y}}(f_{yx}(f_{\tilde{y}y}(\tilde{y})))$ forces the mappings to be less diverse? Do the authors have any explanation about this observation?

In sum, the proposed MPDL is reasonable and seems useful in the conducted experiments. The paper is clear. The experiments can be improved in order to better convince the readers about the model's effectiveness.

**Time Spent Reviewing:**

10

---

> ### Author Response · Authors · 2021-08-10
> **Response to Reviewer FhxC**
>
> Thank you for the valuable comments that help us improve the work. Below we address the concerns mentioned in the review:
>
> *Q1: An intuitive baseline can be a pipelined method, where the first model is to generate the response in style S_0 and the second model is to transfer it into the response in style S_1 in a supervised manner. However, this baseline is not compared in the experiments.*
>
> A1: Thanks for pointing it out. We train the pipelined baseline on TCFC dataset, and the results are shown as follows:
>
> | Model |BLEU-1 | BLEU-2 | Distinct | BERT | SVM |
> |:----:|:----:|:----:|:----:|:----:|:----:|
> | MPDL | 16.5 | 2.07 | 51.3 | 98.6 | 97.1 |
> | Pipeline | 12.9 | 1.33 | 43.1 | 94.5 | 92.7 |
>
> MPDL significantly outperforms the supervised pipelined method. We will add the results in the revision.
>
> &nbsp;
>
> *Q2: In the proposed MPDL framework, it may be possible to use the unpaired D_tra for learning fy~y and f yy~ and add a discriminator to check the quality of y. Hence, the paired style transfer data may not be required, making the target scenarios more practical. Have the authors tried this setting? (Comparing the results of this setting with S2S+BT is fair.)*
>
> A2:  Following your suggestions, we make preliminary exploration along this direction. Denote the data in style S0 and style S1 as $U_0$ and $U_1$, which is non-parallel. Following the techniques from unsupervised NMT, the style transfer model $f_{y\tilde{y}}$ and $f_{\tilde{y}y}$ are implemented as follows:
>
> $\max_{f_{y\tilde{y}}}\frac{1}{|U_1|}\sum_{\tilde{y}\in U_1}\log P(\tilde{y}|\texttt{SG}(f_{\tilde{y}y}(\tilde{y})); f_{y\tilde{y}})$
>
> $\max_{f_{\tilde{y}y}}\frac{1}{|U_0|}\sum_{y\in U_0}\log P(y|\texttt{SG}(f_{y\tilde{y}}(y)); f_{\tilde{y}y})$
>
> where SG(…) means "stop gradient" during propagation. A discriminator to check the quality of y is also used. Denote this method as MPDL_Unpair. The results are shown as follows:
>
> | Model |BLEU-1 | BLEU-2 | Distinct | BERT | SVM |
> |:----:|:----:|:----:|:----:|:----:|:----:|
> | MPDL | 16.5 | 2.07 | 51.3 | 98.6 | 97.1 |
> | MPDL_Unpair | 15.2 | 1.76 | 48.5 | 98.7 | 96.9 |
> | Pipeline | 12.9 | 1.33 | 43.1 | 94.5 | 92.7 |
> | S2S+BT | 12.1 | 1.25 | 42.0 | 86.3 | 86.8 |
>
> The model (MPDL_Unpair) using non-parallel style data performs significantly better than the baselines like Pipeline and S2S+BT. The non-parallel text transfer resources are easy to obtain. We will continue to explore how to accelerate the model and make it more practical.
>
> &nbsp;
>
> *Q3: it is also useful to show the impact of the size of D_tra.*
>
> A3: Thanks for your suggestions. We conduct experiments on TCFC dataset with different sizes of $D_{tra}$. The formal responses results are as follows:
>
> | Model with Different D_tra |BLEU-1 | BLEU-2 | Distinct | BERT | SVM |
> |:----:|:----:|:----:|:----:|:----:|:----:|
> | SRJT_Zero | 15.1 | 1.71 | 43.4 | 97.3 | 96.1 |
> | MPDL_ALL(104k) | 16.5 | 2.07 | 51.3 | 98.6 | 97.1 |
> | MPDL_80k | 15.7 | 1.80 | 48.2 | 98.7 | 97.9 |
> | MPDL_60k | 15.6 | 1.78 | 49.0 | 98.1 | 97.5 |
> | MPDL_40k | 15.3 | 1.83 | 50.3 | 98.4 | 97.2 |
> | MPDL_20k | 14.8 | 1.38 | 48.6 | 98.5 | 97.7 |
>
> There is a positive correlation between the size of $D_{tra}$ and BLEU. When MPDL uses 40k $D_{tra}$, the performance of MPDL has surpassed the baseline SRJT in all metrics.
>
> &nbsp;
>
> *Q4: Are the numbers shown in Table 3 for the TCFC dev set? Because the numbers are quite different from ones in Table 1, can they be compared directly?*
>
> A4: No, they are the average test scores of style S0 and style S1, as we stated in the caption. Therefore, they can be compared directly.
>
> &nbsp;
>
> *Q5: For ablation study, removing discriminators shows the degraded performance in Table 3. Can the authors show the results of removing only one discriminator (either Dx or Dy~), so that we can see which one contributes more?*
>
> A5: Thanks for your suggestions. The results are shown as follows:
>
> | Domain | Model |BLEU-1 | BLEU-2 | Distinct | BERT | SVM |
> |:----:|:----:|:----:|:----:|:----:|:----:|:----:|
> | Formal | MPDL | 16.5 | 2.07 | 51.3 | 98.6 | 97.1 |
> | Formal | w/o D_x | 15.6 | 1.72 | 47.0 | 98.9 | 96.6|
> | Formal | w/o D_y~ | 15.7 | 1.76 | 50.4 | 97.9 | 95.7|
> |Informal | MPDL | 6.92 | 0.69 | 49.3 | 70.3 | 60.7 |
> |Informal | w/o D_x  | 6.78 | 0.63 | 43.6 | 68.9 | 60.6 |
> |Informal | w/o D_y~ | 6.32 | 0.47 | 44.5 | 70.1 | 59.4 |
>
> We find that removing any discriminator hurts the performances.  Both of them contribute to our task.
>
> (1)With D_y~, the generated sequence should match style S1 (i.e., the target style). BERT and SVM scores measure how a style of a sequence matches the target domain, where a BERT classifier and an SVM classifier are applied, respectively.
>
> (2)With D_x, the quality of the pseudo context data x' is evaluated, without which the BLEU-1 and BLEU-2 drop in "formal style" generation.
> We will add them to our paper.
>
> &nbsp;
>
> *Q6: The diversity (Distinct) is sometimes lower in the proposed method. Is it because fyx and fxy~ are not 1-1 mappings and the reconstruction forces the mappings to be less diverse? Do the authors have any explanation about this observation?*
>
> A6: Yes, that's one reason. Another reason is that diversity heavily depends on the amount of data from the target domain.  The TCFC has 500k non-parallel formal texts, and the SDGC has 18.4k non-parallel Shakespearean texts. MPDL can produce better diversity in low-resource scenarios. Therefore, the diversity performance in SDGC is better than that in TCFC.

---

> > ### Comment · Reviewer_FhxC · 2021-08-19
> > **Raised score to 7**
> >
> > Thanks a lot for addressing my concerns.
> > The additional experiments look valuable.
> > Hence, I have raised the score to 7.

---

### Official Review · Reviewer_nC2K · 2021-07-15

**Rating:** 6
**Confidence:** 3

**Summary:**

The authors propose a method for learning stylized dialogue response generation: given a context, generate a response conditioned on a particular style domain (e.g. modern vs. shakespearean english). The method, Multi-Pass Dual Learning (MPDL), leverages large corpora of non-parallel data in tandem. Specifically, the framework involves several forward and backward models between the context and two styles (one which has labeled data from the context to the style, and one that does not); the models produce intermediate data that is used to train the desired style generation model. To make sure that the intermediate data is being used appropriately, the authors introduce two discriminators that evaluate the quality of the data and determine whether it falls in the respective style/domain. The scores of these discriminators weight the importance of the loss computed on that data to ensure that higher-quality pseudo tuples are more impactful in the learning process. The authors evaluate the framework on two style-transfer datasets, and show that the method outperforms 5 baselines, achieving state-of-the-art results in automated metrics and outperforming nearly all models in human evaluations. A case study and ablations are presented as well that highlight and support the authors’ design decisions.

**Limitations And Societal Impact:**

- I have several questions for the authors; in general, the presented equations all made sense, but required that I re-read them several times to resolve the notation.
    - On line 56, the third contribution is stated that they provide a new dataset for this task; it is unclear what dataset they have provided (is it just the sampled set of data?)
    - The derivation of Pr(y\~|x, f_xy\~) in lines 112-113 is not super clear to me (why is y~ only related to x)? (similarly for lines 114-115)
- The scores presented in figure 2 do not seem to align with the quality of the input/output sentences (are they swapped?)
- I believe the first line of equation 7 has a typo, which confused me for quite a bit: should the last part read log(1-Dx(Hf(x)), rather than Hf(y)?
- Where does the informal response in “(context, informal response) dialogue pairs” (line 176) come from?
- What does TCFC stand for?
- How does the model seem to have higher performance on the style classifiers than human responses for the formal response target in the TCFC task?
- Why does SFusion have a much higher Distinct score for the TCFC dataset than it does for SDGC (such that MPDL is now better)?
- I think a qualitative case study of the shakespeare data would be interesting to evaluate the qualitative performance of the model


**Main Review:**

*Originality*: The authors build upon the concept of dual learning and apply it to dialogue response generation; the framework is novel in the sense that the dual learning is required for three domain mappings, and is additionally novel in its use of discriminators to measure the quality of the intermediate data; related work is generally adequately cited, though there are a few cites that are not included (e.g. [1] https://arxiv.org/abs/2009.10855). In general, it is the first paper I have seen applying such a method to dialogue.

*Quality*: The claims are generally well supported, with the presented results clearly identifying MPDL as the preferred method amongst several interesting baselines. Relevant ablations and case studies are considered that highlight the impact of various components in the framework (e.g., the multi-pass and discriminator components are important indeed). Some of the results are not clearly justified or explained, however; see limitations below.

*Clarity*: The theoretical derivations are presented cleanly and concisely; although several pieces of notation are introduced, the proposed method and loss functions are all generally understandable. The results tables and figures are very clearly presented as well. However, I had several questions regarding the presentation of the overall method; see limitations below.

*Significance*: The method is understandable and reproducible and appears to bring strong gains to style generation tasks. What is especially interesting and important is the way that the method can leverage unlabeled data, which makes the method quite generalizable (which is supported by its use on two completely different style-transfer domains).


[1] Smith et. al., "Controlling Style in Generated Dialogue", https://arxiv.org/abs/2009.10855

**Time Spent Reviewing:**

1.75 hours

---

> ### Author Response · Authors · 2021-08-10
> **Response to Reviewer nC2K**
>
> Thank you for the valuable comments that help us improve the work. We will refine the notation details and improve the cites in the revision  (e.g. [1] https://arxiv.org/abs/2009.10855). Below we address the concerns mentioned in the review:
>
> *Q1: On line 56, the third contribution is stated that they provide a new dataset for this task; it is unclear what dataset they have provided (is it just the sampled set of data?)*
>
> A1: Our provided dataset, Shakespearean Dialogue Generation Corpus (SDGC), is a new test set with 2000 data pairs. The dataset is human-annotated.
>
> 1.There are 1000 dialogue pairs, where both the context and the response are modern English.
>
> 2.There are 1000 dialogue pairs, where the context is modern English, and the target is the reply of Shakespearean styles.
>
> &nbsp;
>
> *Q2: The derivation of Pr(y~|x, f_xy~) in lines 112-113 is not super clear to me (why is y~ only related to x)? (similarly for lines 114-115)*
>
> A2: In $P(\tilde{y}^\prime|y,x,\tilde{y};f_{\tilde{y}y}, f_{yx}, f_{x\tilde{y}})$, we want to calculate the probability of $\tilde{y}^\prime$, given the conditions $y,x,\tilde{y}$. $\tilde{y}^\prime$ is reconstructed by the model $f_{x\tilde{y}}$ with $x$ as input, and the remaining elements are not related to this reconstruction process.
>
> &nbsp;
>
> *Q3: The scores presented in figure 2 do not seem to align with the quality of the input/output sentences (are they swapped?)*
>
> A3: No, they are not swapped.
> $D_x$(sentence) is used to verify whether the input sentence belongs to the context domain. Our model outputs a 0.45 score for the sentence “I'm so excited for Brazil to come out.”, which shows that the sentence is OK as a dialogue input since “Brazil” in the context and the “Brazilian” in the response $\tilde{y}$ are consistent.
>
> $D_\tilde{y}$ is used to verify whether the input belongs to the domain of style 1, and in this example, the formal response. “I’m gonna” is an informal representation, and our model outputs a score 0.13, which shows that the sentence is not quite formal.
>
> &nbsp;
>
> *Q4: I believe the first line of equation 7 has a typo, which confused me for quite a bit: should the last part read log(1-Dx(Hf(x)), rather than Hf(y)?*
>
> A4: Yes, this is a typo. Thanks for your suggestions, and we will revise them accordingly.
> In the first row of Equation (7), the last part should be $H^{S0}_b(y)$.
> In the second row of Equation (7), the last part should be $H^{S1}_f(y)$.
>
> The reason is: $H^{S0}_b(y)$ is used to generate pseudo x' for inverse dialogue response generation task. $H^{S1}_f(y)$ is used to generate pseudo $\tilde{y}'$ for text style transfer task. They are false samples generated by the generator in GAN. We will add more details in the revision.
>
> &nbsp;
>
> *Q5: Where does the informal response in “(context, informal response) dialogue pairs” (line 176) come from?*
>
> A5: The dialogue pairs come from TCFC (https://doi.org/10.1609/aaai.v34i05.6468), which are publicly available. For the TCFC, Wu et al. construct the dataset by crawling message-response pairs from Twitter.
>
> &nbsp;
>
> *Q6: What does TCFC stand for?*
>
> A6: TCFC stands for “Twitter Conversation Formality Corpus”(https://doi.org/10.1609/aaai.v34i05.6468). TCFC contains the parallel data between context and informal style, the parallel data between informal style to formal style, and non-parallel formal sentences.
>
> &nbsp;
>
> *Q7: How does the model seem to have higher performance on the style classifiers than human responses for the formal response target in the TCFC task?*
>
> A7: Because the classifiers are trained with the GYAFC dataset. GYAFC is also used as the text transfer task D_tra of MPDL.  Human responses are completely new data for the classifiers.
>
> &nbsp;
>
> *Q8: Why does SFusion have a much higher Distinct score for the TCFC dataset than it does for SDGC (such that MPDL is now better)?*
>
> A8:  As we stated in Line 209 to Line 211, the distinct score measures the diversity of the generated sequences, which is the ratio between the number of unique n-grams and the number of all n-grams in the generated responses. SFusion learns the patterns from the data of the Style S1 domain (i.e., the styles we want to learn), and more data is helpful to increase the diversity. TCFC contains 500k non-parallel data of Style S1 domain while SDGC has only 18.4k. Therefore, SFusion has a better Distinct Score.
>
> &nbsp;
>
> *Q9: I think a qualitative case study of the shakespeare data would be interesting to evaluate the qualitative performance of the model.*
>
> A9: In Table 7 of the Appendix, we show some cases with Shakespeare styles in Table 7 and provide some discussions from Line 26 to Line 29. We will add more cases in the revision.

---

### Official Review · Reviewer_vM8L · 2021-07-16

**Rating:** 7
**Confidence:** 4

**Summary:**

This paper studies the stylized dialog generation problem, in which the problem setup contains contexts C, responses with the original style S0, and responses with the desired style S1. The contributions of this paper includes:
- A novel multiple-pass dual learning framework that explores the interaction between the pairwise relationship among C, S0 and S1.
- The authors propose an additional discrimination to determine the text quality generated by the backward models, which has been shown to be helpful for the task.
- Evaluations are compared to state-of-the-arts on two large text style transfer datasets, and show good amounts of improvements on both automatic and manual metrics.
- Ablation studies further demonstrated the necessity of each component, especially convinced the capacity of discriminator (which is pretty clear from Figure 3).
- The authors also release their code and data.

**Limitations And Societal Impact:**

There are a few flaws in the evaluation part:
- Instead of reporting BLEU-1 and BLEU-2, it's better to report BLEU-4, which is much widely adopted.
- The author could consider to increase the number of samples for human eval to increase the evaluation significance. Also, it would be good to disclose the variance of manual's judgements in each metric.

**Main Review:**

- Originality: There have been many work exploring the stylized dialog generation problem without parallel data. But it's new for this paper to use dual learning to explore the interactions between the pairwise relationship between context C, source style S0 and target style S1. The proposed discriminator component is also intuitive and effective (verified by the experiments), though I am wondering if pretraining such discriminator component might be more effective than optimizing it jointly with the dialog generation loss. Overall, the method is novel and technically sound.
- Quality & Clarity: In general, the paper's writing is clear and easy to follow. There are a few questions to me still: 1) In Equation 9, the optimization loss is defined as the sum of losses from the four components, which gave me the impression that the sum of losses is jointly optimized. However, from Algorithm 1 in the Appendix section, the different components are optimized sequentially. I would love to see the confusion could be resolved. 2) In Table 2, the style consistency scores by auto metrics (e.g., BERT and SVM) don't seem correlate well with the manual judgements, when you compare MPDL with S2S+BT. Is that because the small samples in manual testing, or inaccurate consistency prediction by BERT or SVM?
- Significance: Evaluations are compared to state-of-the-arts on two large text style transfer datasets, and demonstrate significant improvements over strong baselines on a subset of metrics. Human evaluation confirmed that the model generates more style-constrained responses. Ablation studies also provide insights about why this method work.

**Time Spent Reviewing:**

3

---

> ### Author Response · Authors · 2021-08-09
> **Response to Reviewer vM8L**
>
> Thank you for the valuable comments that help us improve the work. Below we address the concerns mentioned in the review:
>
> *Q1: I am wondering if pretraining such discriminator component might be more effective than optimizing it jointly with the dialog generation loss.*
>
> A1: Following your suggestion, we conduct experiments on TCFC dataset. We pretraining the discriminator for 8000 steps, and then keep it fixed (PreD). The results are as follows:
>
> | Domain | Model |BLEU-1 | BLEU-2 | Distinct | BERT | SVM |
> |:----:|:----:|:----:|:----:|:----:|:----:|:----:|
> | Formal | MPDL | 16.5 | 2.07 | 51.3 | 98.6 | 97.1 |
> | Formal | PreD | 16.2 | 2.01 | 50.4 | 98.2 | 97.1 |
> |Informal | MPDL | 6.92 | 0.69 | 49.3 | 70.3 | 60.7 |
> |Informal | PreD | 6.60 | 0.58 | 47.2 | 70.7 | 60.2 |
>
> The performance of pretraining discriminators is not as good as joint training. We will add the results in the revision.
>
> &nbsp;
>
> *Q2: In Equation 9, the optimization loss is defined as the sum of losses from the four components, which gave me the impression that the sum of losses is jointly optimized ...*
>
> A2: Yes, the optimization loss is jointly optimized. We will resolve the confusion about Algorithm 1 in the revision.
>
> &nbsp;
>
> *Q3: In Table 2, the style consistency scores by auto metrics (e.g., BERT and SVM) don't seem correlate well with the manual judgements, when you compare MPDL with S2S+BT.*
>
> A3: During the manual judgements, due to subjective reasons and the small number of samples, the Style-Con. score of S2S+BT is only close to the second level, resulting in inconsistency with automatic evaluation. We will consider increasing the number of manual evaluations, and will disclose the variance of manual judgements in each metric in the revision.
>
> &nbsp;
>
> *Q4: Instead of reporting BLEU-1 and BLEU-2, it's better to report BLEU-4, which is much widely adopted.*
>
> A4: Following your suggestion, the BLEU-4 scores on TCFC are shown as follows:
>
> | Domain | Model |BLEU-4 |
> |:----:|:----:|:----:|
> | Formal | MPDL | 0.16 |
> | Formal | SRJT | 0.08 |
> |Informal | MPDL | 0.09 |
> |Informal | SRJT | 0.02 |
>
> We can see that our method is still the best one in terms of BLEU-4.
>
> &nbsp;
>
> *Q5: The author could consider to increase the number of samples for human eval to increase the evaluation significance. Also, it would be good to disclose the variance of manual's judgements in each metric.*
>
> A5: Thank you for pointing this out. We will increase the  samples for human evaluation in the future.
> For each metric, we will give the variance of TCFC dataset in formal responses as follows:
>
> | Model | Fluency | Relevance | Style-Con. |
> |:----:|:----:|:----:|:----:|
> | SFusion| 0.027 | 0.017 | 0.020 |
> | SRJT | 0.017 | 0.012 | 0.016 |
> |MPDL | 0.015 | 0.011 | 0.015 |
> |Human | 0.006 | 0.005 | 0.013 |
>
> Compared with SFusion and SRJT, the evaluation metrics of our proposed MPDL have smaller variance.

---

### Decision · Program_Chairs · 2021-09-27

**Decision:**

Accept (Poster)

**Comment:**

This paper proposes a new method for stylized dialog generation. The task is that one learns a model to automatically generates a stylized response given a context in dialogue. The authors apply the technique of dual learning to address the problem by learning bi-directional transformations of texts between the contexts C, the original responses S0, and responses with the desired style S1.

Pros
* The paper is generally clearly written.
* A new approach has been proposed, which appears to be reasonable and technically sound.
* The proposed approach appears to be new.
* Experimental results have shown that the proposed method outperforms the baselines on two large datasets. An ablation study has also been conducted.
* The authors also plan to release their code and data.

Cons
* There are some issues with the presentation. The authors have addressed some of them in the rebuttal. They are encouraged to reflect the discussion results in the new version of the paper.
* The authors also conducted additional experiments during the rebuttal. They are asked to include the results in the new version of the paper.
* The proposed method appears to be complicated. One may doubt whether it can be really used in practice.*